# Jointly looking to the past and the future in visual working memory

**Baiwei Liu\*, Zampeta-Sofia Alexopoulou, Freek van Ede\***

Institute for Brain and Behavior Amsterdam, Department of Experimental and Applied Psychology, Vrije Universiteit Amsterdam, Amsterdam, Netherlands

**Abstract** Working memory enables us to bridge past sensory information to upcoming future behaviour. Accordingly, by its very nature, working memory is concerned with two components: the past and the future. Yet, in conventional laboratory tasks, these two components are often conflated, such as when sensory information in working memory is encoded and tested at the same location. We developed a task in which we dissociated the past (encoded location) and future (to-be-tested location) attributes of visual contents in working memory. This enabled us to independently track the utilisation of past and future memory attributes through gaze, as observed during mnemonic selection. Our results reveal the joint consideration of past and future locations. This was prevalent even at the single-trial level of individual saccades that were jointly biased to the past and future. This uncovers the rich nature of working memory representations, whereby both past and future memory attributes are retained and can be accessed together when memory contents become relevant for behaviour.

**\*For correspondence:**
b.liu@vu.nl (BL);
freek.van.ede@vu.nl (FvE)

**Competing interest:** The authors declare that no competing interests exist.

## eLife assessment

This **important** study advances our understanding of how past and future information is jointly considered in visual working memory by studying gaze biases in a memory task that dissociates the locations during encoding and memory tests. The evidence supporting the conclusions is **convincing**, with state-of-the-art gaze analyses that build on a recent series of experiments introduced by the authors. This work will be of broad interest to vision scientists interested in the interplay of vision, eye movements, and memory.

## Introduction

Working memory is a fundamental cognitive function that enables us to hold onto past sensory information in service of upcoming future behaviour (***D'Esposito and Postle, 2015***; ***van Ede and Nobre, 2023***; ***Rainer et al., 1999***). Accordingly, by its very nature, working memory is concerned with two components: the past and the future.

In conventional laboratory tasks, past and future components are often conflated such as when sensory information in working memory is tested at the same location as where it was encoded. By contrast, in the dynamic situations we face every day, sensory information often disappears at one specific location (where it enters visual working memory) but becomes relevant at another location (as also in ***Brincat et al., 2021***; ***Doherty et al., 2005***; ***Woodman et al., 2012***; ***Kahneman et al., 1992***; ***Zaksas et al., 2001***). Imagine trying to capture a photograph of a precious bird species that you just saw disappear behind a building. Your working memory of the bird is likely to consider not only where you last saw the bird, but also where you expect it to re-appear to capture it on camera. Such situations raise an interesting, underexplored question: when past and future locations of memory

**Figure 1.** Directional gaze biases by past and future locations during mnemonic selection overlap in time. (**a**) Task schematic. Participants memorised two oriented gratings with different colours presented either vertically or horizontally. Following a delay, a colour change of the central fixation dot prompted participants to select the colour-matching item from working memory to report its orientation later. After another delay, two test gratings appeared transiently and participants compared the cued memory grating to the relevant test grating (clockwise/counter-clockwise judgement) that was determined by the 'future rule' that was stable within each block. After a response, feedback (0: wrong, 1: correct) was presented at the side of the relevant test grating. Dash lines serve to explain the association between the encoding and test locations and were never presented in the actual experiment. (**b–c**) Time courses of gaze shift rates (number of saccades per second) for shifts toward and away from the encoded (panel b) and to-be-tested (panel c) locations. (**d**) Overlays and comparisons of the difference in gaze-shift rates (toward minus away) for the past (encoded) location and the future (to-be-tested) location. Horizontal lines indicate significant temporal clusters (cluster-based permutation test, p<0.001). Data are presented as mean values with shading reflecting 95% confidence intervals, calculated across participants (n=25).

The online version of this article includes the following figure supplement(s) for figure 1:

**Figure supplement 1.** Four possible associations between encoding and test locations.

**Figure supplement 2.** Early saccade biases by past and future memory attributes are predominantly driven by microsaccades.

**Figure supplement 3.** Trials with vertical or horizontal configurations show similar joint consideration of past and future memory attributes.

**Figure supplement 4.** Gaze biases in an extended time window as a complement to *Figure 1* and *Figure 1—figure supplement 2*.

contents are not the same, does the brain code internal representations with regard to past, future, or both?

To address this question, we developed a task in which we dissociated the encoding (past) and to-be-tested (future) locations associated with visual representations in working memory. This enabled us to experimentally isolate past and future memory attributes and to track their respective utilisation through spatial biases in gaze behaviour in healthy human volunteers.

## Results

Twenty-five human volunteers performed a working-memory task in which visual memory items were encoded and tested at different locations (*Figure 1a*). Participants memorised two coloured gratings with different orientations presented either vertically or horizontally. The crucial manipulation was that we always tested memory content in the orthogonal axis and at a predictable location (depending

on the future rule that we varied across sessions; see *Figure 1—figure supplement 1* for the four possible rules).

After a delay period, we cued the relevant memory item by changing the colour of the central fixation dot. At this stage, participants were required to select the colour-matching grating from working memory in order to compare it to the upcoming test stimulus (clockwise/counter-clockwise judgement). Crucially, we always presented two stimuli at the test-phase of which only one was relevant, as determined by the future rule. For example, in *Figure 1a*, after the green memory item is cued, the relevant test stimulus will be the right stimulus, given the future rule (top item tested on right) in this session. Note how the colour cue only ever informed the to-be-tested memory content but never directly informed the to-be-tested future location. The to-be-tested location was an attribute of the cued memory content but not of the cue itself.

Participants were able to perform this dynamic visual working-memory task, with an average accuracy of 70±2percent correct (mean ± SEM) and an average reaction time of 1218±125 ms.

## Gaze reveals the use of both past and future memory attributes that are considered at overlapping time windows

To track the utilisation of past and/or future locations associated with working memory contents, we tracked spatial biases in gaze following the cue to select either memory item. Specifically, we focused on directional biases in saccades, that have previously been shown to be sensitive to selective spatial attention (*Engbert and Kliegl, 2003*; *Hafed and Clark, 2002*; *Rolfs, 2009*; *Laubrock et al., 2010*; *Yuval-Greenberg et al., 2014*; *Shelchkova and Poletti, 2020*; *Fernández et al., 2023*; *Lowet et al., 2018*), even when directed internally (*Liu et al., 2022a*; *van Ede et al., 2021*; *van Ede et al., 2019*; *van Ede et al., 2020*).

As shown in *Figure 1b*, after cue onset, saccades became biased in the direction of the encoded (past) location of the selected memorized target, as demonstrated by significantly more gaze shifts toward vs. away from the encoded location of the target (*Figure 1b*; cluster p<0.001), starting from around 200 ms after the cue. This is consistent with our prior demonstrations of directional eye-movement biases within the spatial layout of working memory (*Liu et al., 2022a*; *van Ede et al., 2021*; *van Ede et al., 2019*; *van Ede et al., 2020*). In our current task, this uniquely reveals how the retention of items in working memory continues to rely on their past encoded locations, even if items are known to become relevant (tested) at another location.

Having established that past (encoded) memory locations were still utilised by participants in our task (despite us never asking about memory-item locations), an interesting question becomes when the future memory attribute (to-be-tested location) would be considered after the selection cue. Intuitively, participants may select the relevant item at its past location, before considering the relevant future test location – which would yield a serial pattern of past-before-future. In contrast, as shown in *Figure 1c and d*, we found a similar saccade bias to the relevant future location (*Figure 1c*; cluster p<0.001). Strikingly, this future bias also emerged early after the cue. An overlay of the spatial biases (toward vs. away) in the orthogonally manipulated past and future axes (*Figure 1d*), revealed consideration of past and future locations at overlapping time windows. Gaze biases in both axes were driven predominantly by microsaccades (*Figure 1—figure supplement 2*) and occurred similarly in horizontal-to-vertical and vertical-to-horizontal trials (*Figure 1—figure supplement 3*). Moreover, while the past bias was relatively transient, the future bias continued to increase in anticipation of the test stimulus and increasingly incorporated eye movements beyond the microsaccade range (see *Figure 1—figure supplement 4* for a more extended time range).

These data thus suggest the joint consideration – or 'activation' – of past and future memory attributes, at least when analysing past and future memory attributes separately. Below, we provide additional single-trial (single-saccade) evidence for this interpretation.

## Individual saccades reveal truly joint consideration of past and future memory attributes

In principle, the observed joint activation of the past and future locations associated with the cued memory content in the trial-averaged and participant-averaged data could result from two alternative scenarios with different interpretations. First, either the past or the future alone may be considered in different trials and/or participants, without past and future memory attributes ever being considered

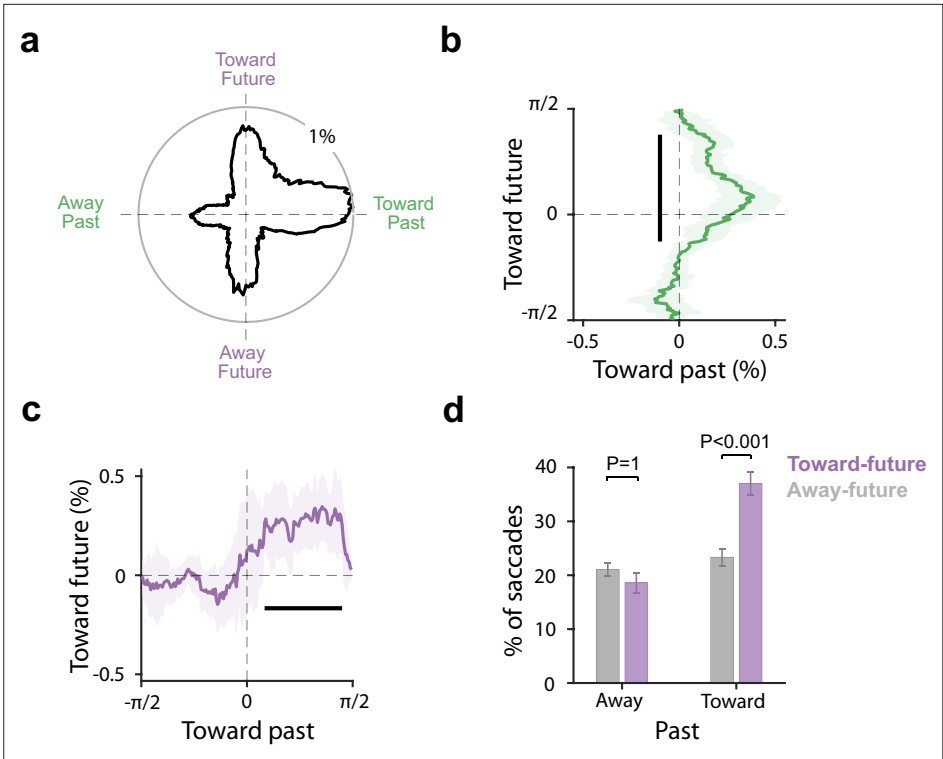

**Figure 2.** Individual saccades are jointly biased to past and future memory attributes. (**a**) The distribution of the direction of the first saccades we detected after cue onset relative to past (horizontal) and future (vertical) locations. Data from the different sessions were rotated to match a common coordinate frame. (**b**) The bias toward the past (x-axis) as a function of saccade direction with regard to the future (y-axis). (**c**) The bias toward the future (y-axis) as a function of saccade direction with regard to the past (x-axis). In (**b–c**), The bold black line indicates the significant temporal cluster (cluster-based permutation test, p<0.001). Data are presented as mean values with shading indicating 95% confidence intervals, calculated across participants (n=25). (**d**) The percentage of identified first saccades toward or away from the future location as a function of whether the same saccades were also biased toward or away from the past location. Error bars in panel d indicate ±1 SEM calculated across participants (n=25).

The online version of this article includes the following figure supplement(s) for figure 2:

**Figure supplement 1.** Distribution of saccade directions relative to the future rule from encoding onset.

together. Alternatively, participants may truly consider both past and future memory attributes jointly at the single-trial, single-saccade level.

While it is notoriously hard to disentangle the single-trial interpretation of averaged data (*Stokes and Spaak, 2016*), we were here able to do so by interrogating the individual saccade characteristics for which the two alternative scenarios make different predictions. In the first scenario, saccades should be biased to either the past or the future location, but there should be no dependency between them (i.e., a saccade may be biased to the past location regardless of the future location, and vice versa). In contrast, in the second scenario, with truly joint consideration of the past and future, there should be a clear dependency: it should be those saccades that are biased to the past that are also biased to the future. In other words, the future-biased saccades should predominantly be driven by the past-biased saccades, and vice versa.

To disentangle these alternatives at the single-trial level of individual saccades, we focused on the first saccades after the cue, in the 200–600 ms window. We previously identified these windows as the relevant window for microsaccade biases by internal selective attention (*Liu et al., 2022a*), and this is also where we observed the overlapping past and future biases in the average here (see *Figure 1d*). To facilitate visualisation and quantification, we rotated all detected saccades to match a consistent coordinate frame, in which the past location is represented horizontally (right = towards, left = away), and the future location vertically (top = toward, bottom = away; see *Figure 2a*).

As shown in *Figure 2a*, the majority of first-detected microsaccades in our window of interest were made toward the past (right >left) and toward the future (top >bottom), replicating our prior analyses. Critically, this visualisation and quantification enabled us to disentangle the two alternatives sketched above in which past and future saccades at the single-trial (individual-saccade) level were either independent (past bias regardless of future and future bias regardless of past) or dependent (joint past and future bias). Our data supported the latter.

*Figure 2b* shows the bias toward the past as a function of saccade direction with regard to the future. Likewise, *Figure 2c*, shows the future bias as a function of whether saccades were also biased to the past. In both cases, we see a clear dependency: the past bias is particularly pronounced for saccades that also have a future bias (*Figure 2b*) and, vice versa, the future bias is most pronounced for saccades that also have a past bias (*Figure 2c*). This is perhaps best appreciated by the binarised quantification of these same data in *Figure 2d*: showing the percentage of identified first saccades toward or away from the future location, as a function of whether the same saccades were also biased toward or away from the past location. We found a clear interaction ($F(1, 24)=18.1$, $p<0.001$, partial $\eta^2=0.43$), whereby the bias toward the future location was exclusively observed for those saccades that were also biased toward the past location. Indeed, follow-up t-tests revealed no difference in future bias for saccades that were away from the past ($t(24) = 1.13$, $P_{Bonferroni}=1$, $d=0.23$), but a clear future effect for saccades that were toward the past ($t(24)=4.65$, $P_{Bonferroni}<0.001$, $d=0.93$). This would not be expected if single saccades cared exclusively about either the past or the future location. Instead, this provides single-trial-level support with the truly joint consideration of past and future memory attributes.

## Discussion

Here, we brought the study of visual working memory into a dynamic context by experimentally dissociating (orthogonalising) the past (last-seen location) and future (to-be-tested location) attributes of visual memory contents, and independently tracking the utilisation of these two attributes through the gaze. Doing so, we unveil a novel, fundamental property of working memory – the joint availability and utilisation (i.e. selection) of past and future memory attributes. As such, our data provide key support for the proposal that memory is fundamentally future-oriented (*van Ede and Nobre, 2023*; *Rainer et al., 1999*; *Nobre and Stokes, 2019*; *Schacter et al., 2007*), while also reminding us that past memory attributes are not forgotten, even when these do not become relevant again.

Our finding of joint utilisation of past and future memory attributes emerged from at least two alternative scenarios of how the brain may deal with dynamic everyday working memory demands in which memory content is encoded at one location but needed at another. First, memory contents could have directly been remapped (*Brincat et al., 2021*; *de Vries and van Ede, 2024*; *Hayhoe and Ballard, 2005*; *Pelz and Canosa, 2001*) to their future-relevant location. However, in this case, one may have expected to exclusively find a future-directed gaze bias, unlike what we observed. Moreover, using a complementary analysis of saccade directions along the axis of the future rule (*de Vries and van Ede, 2024*), we found no direct evidence for remapping in the period between encoding and cue (*Figure 2—figure supplement 1*). Second, when dealing with multiple memory contents, contents could be stored at the past location at first and the future location could be considered only after relevant memory content has been selected (in which case the past bias should have *preceded* the future bias). In contrast, our data suggest that the brain simultaneously retains the copy of both past and future-relevant locations in working memory, and (re)activates each during mnemonic selection. Thus, while it is not surprising that the future location is considered (*Hayhoe and Ballard, 2005*; *Pelz and Canosa, 2001*; *Mennie et al., 2007*), it is far less trivial that both past and future attributes would be retained and (re)activated together. This is our central contribution.

By capitalising on the discrete nature of saccades, we were able to demonstrate the truly joint consideration of past and future attributes, at the single-trial level. As such we were able to bypass a fundamental challenge of disentangling multiple potential single-trial interpretations when only having trial-average data available (for related discussions, see *Stokes and Spaak, 2016*; *Jones, 2016*; *Nobre and van Ede, 2020*). For example, when only considering the temporally overlapping past and future signals at the trial-average level, it was impossible to tell whether these joint effects resulted from a mix of trials and/or participants that relied on either the past *or* the future. By considering the discrete events – the individual saccades at the single-trial level – we could demonstrate

how biases to the past and the future co-existed at the single-saccade level, supporting a truly joint consideration of past and future.

In the current study, we tracked spatial attention using microsaccades (*Engbert and Kliegl, 2003*; *Hafed and Clark, 2002*; *Rolfs, 2009*; *Laubrock et al., 2010*; *Yuval-Greenberg et al., 2014*; *Shelchkova and Poletti, 2020*; *Fernández et al., 2023*; *Lowet et al., 2018*; *Liu et al., 2022a*; *van Ede et al., 2021*; *van Ede et al., 2019*; *van Ede et al., 2020*). Compared to the commonly used online indicator of spatial attention, such as electrophysiological measures, microsaccades have important features that make them a promising complementary tool for uncovering the mechanisms of spatial attention (*Rolfs, 2009*; *Engbert, 2006*), including when directed internally as we have shown here. First, as we have discussed above, microsaccades are discrete events, allowing to track spatial attention at the single-trial level. Second, microsaccades are not limited to tracking spatial attention toward the left or right visual field, as electrophysiology indicators often are. These two aspects were paramount to our demonstration of joint single-trial consideration of past and future memory attributes, and are likely to open additional doors for future investigations.

While the past gaze bias that we report here replicates our own prior studies (*Liu et al., 2022a*; *van Ede et al., 2021*; *van Ede et al., 2019*; *van Ede et al., 2020*), here we for the first time demonstrate a similar bias to the future-relevant memory location that is similarly driven by microsaccades. This signal may reflect either of two situations: the selection of a future copy of the cued memory content or anticipatory attention to the anticipated location of its associated test-stimulus. Either way, by the nature of our experimental design, this future signal should be considered a content-specific memory attribute for two reasons. First, the two memory contents were always associated with opposite testing locations, hence the observed bias to the relevant future location must be attributed specifically to the cued memory content. Second, we cued which memory item would become tested based on its colour, but the to-be-tested location was dependent on the item's encoding location, regardless of its colour. Hence, consideration of the item's future-relevant location must have been mediated by selecting the memory item itself, as it could not have proceeded via cue colour directly. Accordingly, our data reveal how this future feature can be accessed from memory together with the specific memory content that it is associated with, implying joint storage and utilisation of past and future memory attributes.

Building on the above, at face value, our task may appear like a study that simply combines two established tasks: tasks using retro-cues to study attention in working memory (e.g. *van Ede and Nobre, 2023*; *Griffin and Nobre, 2003*; *Panichello and Buschman, 2021*; *Souza and Oberauer, 2016*) and tasks using pre-cues to study the orienting of spatial attention to an upcoming external stimulus (e.g. *Griffin and Nobre, 2003*; *Panichello and Buschman, 2021*; *Posner, 1980*; *Worden et al., 2000*; *Schmidt et al., 2002*). A critical difference with common pre-cue studies, however, is that the cue in our task never directly informed the relevant future location. Rather, as also stressed above, the future location was a feature of the cued memory item (according to the future rule), and not of the cue itself. Note how this type of scenario may not be uncommon in everyday life, such as in our opening example of a bird flying behind a building. Here too, the future relevant location is determined by the bird – i.e., the memory content – itself.

In our study, the past location of the memory items was technically irrelevant for the task and could thus, in principle, be dropped after encoding. One possibility is that participants remapped the two memory items to their future locations soon after encoding, and had started – but not finished – dropping the past location by the time the cue arrived. In such a scenario, the past signal is merely a residual trace of the memory items that serves no purpose but still pulls the gaze. Alternatively, however, the past locations may be utilised by the brain to help individuate/separate the two memory items. Moreover, by storing items with regard to multiple spatial frames (*Draschkow et al., 2022*) – here with regard to both past and future visual locations – it is conceivable that memories may become more robust to decay and/or interference. Also, while in our task past locations were never probed, in everyday life it may be useful to remember where you last saw something before it disappeared behind an occluder. In future work, it will prove interesting to systematically vary the delay between encoding and cue to assess whether the reliance on the past location gradually dissipates with time (consistent with dropping an irrelevant feature), or whether the past trace remains preserved despite longer delays (consistent with preserving utility for working memory).

Our data complement other recent studies investigating visual working memory in more dynamic contexts (*Brincat et al., 2021*; *van Ede et al., 2021*; *Draschkow et al., 2022*; *Chung et al., 2022*; *de Vries et al., 2020*; *Xie et al., 2022*), and showcase the rich nature of working memory representations. For example, akin to our finding of joint consideration of past and future locations in working memory, recent studies have uncovered the joint consideration of allocentric and egocentric spatial frames for working memory (*Draschkow et al., 2022*; *Fiehler et al., 2014*). While storing memory content at a single location (or with reference to a single frame) would appear more intuitive and efficient, our data reveal an intriguing alternative. We speculate that the joint retention of multiple spatial attributes may make memories more robust, as well as more flexible for serving continuously evolving demands during everyday behaviour.

## Methods

### Participants

Twenty-five healthy human volunteers participated in the study (age range: 20–27; 10 male and 15 female; 23 right-handed; 10 corrected-to-normal vision: five glasses and five lenses). Sample size of 25 was determined a-priori based on previous publications from the lab with similar experimental designs, and that relied on the same outcome measure (*Liu et al., 2022a*; *van Ede et al., 2021*; *van Ede et al., 2020*). To achieve the intended sample size, three participants were replaced due to chance-level performance.

### Stimuli and procedure

Participants performed a visual working memory task in which we orthogonalised the encoding and to-be-tested location in order to track the utilisation of past and future working-memory attributes.

Participants were required to encode and maintain two visual items (tilted coloured gratings) in working memory to later report one of their orientations (*Figure 1a*). Each trial began with a brief (250 ms) encoding display in which two to-be-memorised gratings with different colours and orientations appeared vertically or horizontally on either side of the fixation dot, at 4degrees visual angle. After a retention delay of 1250 ms, the fixation dot changed colour for 1000 ms. This colour change served as a 100% valid retro-cue, prompting participants to select the colour-matching target memory item. The retro-cue was followed by another retention delay of 500 ms, before the test display. The test display always contained two black gratings with different orientations that were presented vertically or horizontally on either side of the fixation dot (again at four visual degrees to each side). Based on a rule that described in the following paragraph, one of the black gratings was relevant to the task (the test grating), while the other merely served as a filler. After seeing the test display, participants were required to compare the cued memory grating to the relevant test grating and report whether the memory grating should be turned clockwise (using the keyboard button 'j') or counterclockwise (using the keyboard button 'f') rotated to match the relevant test grating in the test display. After responding, feedback ('0' for wrong, or '1' for correct) would be presented for 250 ms at the side of the relevant test grating, indicating the correctness of the response and reinforcing the future rule.

In the encoding display, the gratings were randomly assigned two distinct colours: green (RGB: 133, 194, 18) and purple (RGB: 197, 21, 234) and two distinct orientations ranging from 0° to 180° with a minimum difference of 20° between each other. During the test display, the gratings are always black (RGB: 64, 64, 64). The relevant grating was always rotated 20degrees in either a clockwise or counterclockwise direction compared to the to-be-tested memory grating. The orientation of the irrelevant grating in the test display was chosen randomly.

The unique element of our task was that we dissociated the encoding and testing location by presenting the gratings in the test display on the orthogonal axis as where the items were presented on the encoding display. Relevant examples can be found in *Figure 1a* and *Figure 1—figure supplement 1*. For example, if the two memory items appeared vertically at the top and bottom at encoding, the test gratings would appear horizontally to the left and right in the test display (see *Figure 1a*).

To to-be-tested location was always linked to the encoded location by virtue of a future rule. For counterbalancing reasons, we used four unique rules that were presented across four sessions (*Figure 1—figure supplement 1*) and that always remained stable within a session. In rule 1: the two memory items were encoded vertically on the top and bottom. If the top memory item was cued, the

relevant test grating would be on the right, while if the bottom item would be cued, the relevant test grating would be on the left in the test display. In rule 2, the mapping was reversed: the two memory items would again be encoded vertically, but this time if the top memory item would be cued the relevant test grating is on the left, while if the bottom item is cued the relevant test grating is on the right. Rules 3 and 4 follow the same logic, except that now the memory items are presented horizontally, and the test gratings vertically. Before every session, participants were notified of the future rule that applied to the upcoming session.

To ensure the use of the future rule, we made it task-relevant by always presenting two test gratings in the test display of which only one was relevant: the one that matched the future rule (i.e. without applying the future rule, one would not know which test grating was to be used). It is for this reason that we consider the future location an attribute of the memory items, as each memory item was associated with its own unique test location. Note also how our colour cue never directly informed about the future rule or the to-be-tested location. The cue informed which memory item would become tested based on its colour, but the to-be-tested location was dependent on the cued item's encoding location, regardless of its colour (with item colour and location varying randomly across trials). Accordingly, consideration of the item's future-relevant location must have been mediated by selecting the memory item itself, as it could not proceed via the colour cue directly.

In total, the study consisted of four sessions, each contained five blocks of 32 trials each. At the start of each session, participants were notified of the session-specific future rule and then practiced the task with the current rule for 16 trials before starting the formal session. We did not include practice trials in our analyses. The study lasted approximately 70min per participant.

## Eye-tracking acquisition and pre-processing

Gaze was tracked from a single eye (right eye in all participants except one for which the left eye provided a better signal) using an EyeLink 1000 (SR Research) at a sampling rate of 1000Hz. The eye tracker camera was positioned on the table ~5cm in front of the monitor and ~65cm away from the eyes. Gaze position was tracked continuously along the horizontal and vertical axes. Before recording, the built-in calibration and validation protocols from the EyeLink software were used to calibrate the eye tracker.

After recording, the eye-tracking data were converted from the original.edf to the .asc format and analysed in Matlab using the Fieldtrip analysis toolbox (*Oostenveld et al., 2011*) in combination with custom code. Blinks were marked by detecting 0 clusters in the eye-tracking data. All data from 100 ms before to 100 ms after the detected 0 clusters were set to NaN to eliminate residual blink artefacts. Finally, data were epoched from −1000 to+2000 ms relative to after the onset of the retro-cue.

## Gaze-shift detection

We focused our analysis on spatial biases in gaze shifts (saccades/microsaccades). To identify gaze shifts, we employed a velocity-based method that we established in our prior studies (*Liu et al., 2022a*; *Liu et al., 2022b*), that builds on other velocity-based detection methods (e.g., *Engbert and Kliegl, 2003*). First, gaze velocity was calculated by taking the Euclidean distance between temporally successive gaze-position values in the two-dimensional plane (horizontal and vertical gaze position). Velocity was smoothed with a Gaussian-weighted moving average filter with a 7ms sliding window (using the built-in function 'smoothdata' in MATLAB). When the velocity exceeded a trial-based threshold of five times the median velocity, we marked the first sample after the threshold crossing as the onset of a saccade. To avoid counting the same saccade multiple times, a minimum delay of 100 ms between successive saccades was imposed. Saccade magnitude and direction were calculated by estimating the difference between pre-saccade gaze position (−50–0 ms before saccade onset) vs. the post-saccade gaze position (50–100 ms after saccade onset).

To focus our analysis on (micro)saccades that were driven by attention, we here focused on the 'start microsaccade' defined as the saccade that moves the gaze away from the fixation dot (as opposed to saccades that bring the gaze back to fixation). We extracted start saccades using a method that we recently validated in another study (*Liu et al., 2022b*). For each detected saccade, we estimated the pre- and post-saccade distance from the central fixation dot. If the post-saccade distance was larger than the pre-saccade distance, we defined the saccade as a start microsaccade. To minimise the

contribution of gaze drift, we defined gaze positions associated with looking at the central fixation dot using the median gaze position in the fixation period from [-0.8 to –0.2 ms] relative to cue onset.

Time courses of gaze-shift rates (in Hz) were quantified using a sliding time window of 50 ms, advanced in steps of 1 ms. We additionally decomposed shift rates into a time-magnitude representation (as in *Liu et al., 2022a*), showing the time-resolved rate of attention-driven shifts (toward vs. away) per second, as a function of the saccade size (see *Figure 1—figure supplement 2*). For magnitude sorting, we used successive magnitude bins of 0.2 visual degrees in steps of 0.04 visual degrees.

### Individual-shift level analysis

For our analysis at the single-trial, single-saccade level, we focused on the first start saccade observed during the 200–600 ms post-cue period that we have previously identified as the relevant time window for the effect of interest (*Liu et al., 2022a*). For all 'first saccades,' we then looked at the spatial distribution of saccade directions relative to the relevant past (encoding) and future (to-be-tested) locations associated with the cued memory item. To facilitate visualisation and quantification, we rotated all detected saccades to a common coordinate system, in which the past location was represented horizontally (left=away, right=toward), and the future location vertically (top=toward, bottom = away).

To obtain the proportional distribution of saccades in this coordinate system (see *Figure 2a*), we used successive angular bins of 20degrees in steps of 1 angle degree and simply calculated the proportion of saccades in each angular bin (with angles being defined with reference to past and future locations).

To quantify whether saccades were jointly biased to the past and future (as opposed to either alone), we decomposed these distributions into the bias toward the past (toward vs. away from past) as a function of saccade direction with regard to the future (*Figure 2b*) and the bias toward the future (toward vs. away from future) as a function of saccade direction with regard to the past (*Figure 2c*). Finally, we binarised saccades as either going toward or away from the past and toward or away from the future (*Figure 2d*). This enabled us to test whether saccades were jointly biased to the past and the future (which would predict an interaction whereby those same saccades that were biased to the past were also biased to the future).

### Statistical analysis

To evaluate the reliability of patterns in our gaze data, we used a cluster-based permutation approach (*Maris and Oostenveld, 2007*). This method is ideal for evaluating patterns at multiple neighbouring points while circumventing the problem of multiple comparisons. We used this approach for all evaluations involving a series of data, such as along a time axis (*Figure 1b–d*; *Figure 2b–c*).

To create a permutation distribution, we randomly permuted the trial-average data at the participant level 10,000 times and identified the largest clusters found at each time. The p-values of the clusters in the original data were calculated as the proportion of permutations for which the size of the largest cluster after permutation was larger than the size of the observed cluster in the original, non-permuted data. We created the permutation distribution using Fieldtrip with default cluster settings (grouping adjacent same-signed data points that were significant in a mass univariate t-test at a two-sided alpha level of 0.05 and defining cluster size as the sum of all t-values in a cluster).

In addition, statistical evaluation for the single-trial, single-saccade analysis was performed on the binarised quantification of saccade proportions using a two-way repeated-measures ANOVA with the factors past (toward/away) and future (toward/away). ANOVA results were complemented with Bonferroni-corrected post-hoc t-tests. For measures of effect size, we used Partial eta squared (for our ANOVA) and Cohen's d (for follow-up t-tests). p-values of follow-up t-tests were Bonferroni corrected.

### Acknowledgements

This research was supported by an ERC Starting Grant from the European Research Council (MEMTIC-IPATION, 850636) and an NWO Vidi grant by the Dutch Research Council (grant number 14721) to F.v.E.

# Additional information

## Funding

| Funder | Grant reference number | Author |
| --- | --- | --- |
| European Research Council | 850636 | Freek van Ede |
| Sociale en Geesteswetenschappen, NWO | 14721 | Freek van Ede |

The funders had no role in study design, data collection and interpretation, or the decision to submit the work for publication.

## Author contributions

Baiwei Liu, Conceptualization, Formal analysis, Supervision, Validation, Investigation, Visualization, Writing – original draft, Writing – review and editing; Zampeta-Sofia Alexopoulou, Formal analysis, Investigation, Visualization; Freek van Ede, Conceptualization, Resources, Formal analysis, Supervision, Investigation, Visualization, Writing – original draft, Writing – review and editing, Funding acquisition, Methodology

## Author ORCIDs

Baiwei Liu (ID) http://orcid.org/0000-0002-7857-4456
Zampeta-Sofia Alexopoulou (ID) http://orcid.org/0000-0002-7304-0834
Freek van Ede (ID) http://orcid.org/0000-0002-7434-1751

## Ethics

Experimental procedures were reviewed and approved by the local Ethics Committee at the Vrije Universiteit Amsterdam (reference code: VCWE-2020-155). Each participant provided written consent before participation and was reimbursed €10/hr.

Reviewer #1 (Public Review): https://doi.org/10.7554/eLife.90874.3.sa1
Reviewer #2 (Public Review): https://doi.org/10.7554/eLife.90874.3.sa2
Author response https://doi.org/10.7554/eLife.90874.3.sa3

# Additional files

## Supplementary files

- MDAR checklist

## Data availability

Data and code availability. The corresponding data and code can be found at: https://osf.io/xukzs/ at OSF (https://doi.org/10.17605/OSF.IO/XUKZS).

The following dataset was generated:

| Author(s) | Year | Dataset title | Dataset URL | Database and Identifier |
| --- | --- | --- | --- | --- |
| Liu B, Alexopoulou Z, van Ede F | 2024 | Jointly looking to the past and the future in visual working memory | https://doi.org/10.17605/OSF.IO/XUKZS | Open Science Framework, 10.17605/OSF.IO/XUKZS |

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
