## [Editor Report · eLife assessment]

This **important** study advances our understanding of how past and future information is jointly considered in visual working memory by studying gaze biases in a memory task that dissociates the locations during encoding and memory tests. The evidence supporting the conclusions is **convincing**, with state-of-the-art gaze analyses that build on a recent series of experiments introduced by the authors. This work will be of broad interest to vision scientists interested in the interplay of vision, eye movements, and memory.

---

## [Referee Report · Reviewer #1 (Public Review)]

In this study, the authors offer a fresh perspective on how visual working memory operates. They delve into the link between anticipating future events and retaining previous visual information in memory. To achieve this, the authors build upon their recent series of experiments that investigated the interplay between gaze biases and visual working memory. In this study, they introduce an innovative twist to their fundamental task. Specifically, they disentangle the location where information is initially stored from the location where it will be tested in the future. Participants are tasked with learning a novel rule that dictates how the initial storage location relates to the eventual test location. The authors leverage participants' gaze patterns as an indicator of memory selection. Intriguingly, they observe that microsaccades are directed towards both the past encoding location and the anticipated future test location. This observation is noteworthy for several reasons. Firstly, participants' gaze is biased towards the past encoding location, even though that location lacks relevance to the memory test. Secondly, there's a simultaneous occurrence of an increased gaze bias towards both the past and future locations. To explore this temporal aspect further, the authors conduct a compelling analysis that reveals the joint consideration of past and future locations during memory maintenance. Notably, microsaccades biased towards the future test location also exhibit a bias towards the past encoding location. In summary, the authors present an innovative perspective on the adaptable nature of visual working memory. They illustrate how information relevant to the future is integrated with past information to guide behavior.

---

## [Referee Report · Reviewer #2 (Public Review)]

Summary:

The manuscript by Liu et al. reports a task that is designed to examine the extent to which "past" and "future" information is encoded in working memory that combines a retrocue with rules that indicate the location of an upcoming test probe. An analysis of microsaccades on a fine temporal scale shows the extent to which shifts of attention track the location of the encoded item (past) and the location of the future item (test probe). The location of the encoded grating and test probe were always on orthogonal axes (horizontal, vertical) so that biases in microsaccades could be used to track shifts of attention to one or the other axis (or mixtures of the two). The overall goal here was then to (1) create a methodology that could tease apart memory for the past and future, respectively, (2) to look at the time-course attention to past/future, and (3) to test the extent to which microsaccades might jointly encode past and future memoranda. Finally, some remarks are made about the plausibility of various accounts of working memory encoding/maintenance based on the examination of these time-courses.

Strengths:

This research has several notable strengths. It has a clear statement of its aims, is lucidly presented, and uses a clever experimental design that neatly orthogonalized "past" and "future" as operationalized by the authors. Figure 1b-d shows fairly clearly that saccade directions have an early peak (around 300ms) for the past and a "ramping" up of saccades moving in the forward direction. This seems to be a nice demonstration that the method can measure shifts of attention at a fine temporal resolution and differentiate past from future oriented saccades due to the orthogonal cue approach. The second analysis shown in Figure 2, reveals a dependency in saccade direction such that saccades toward the probe future were more likely also to be toward the encoded location than away from the encoded direction. This suggests saccades are jointly biased by both locations "in memory". The "central contribution" (as the authors characterize it) is that "the brain simultaneously retains the copy of both past and future-relevant locations in working memory, and (re)activates each during mnemonic selection", and that: "... while it is not surprising that the future location is considered, it is far less trivial that both past and future attributes would be retained and (re)activated together. This is our central contribution." The authors provide a nuanced analysis that offers persuasive evidence that past and future representations are jointly maintained in memory.

---

## [Author Response]

The following is the authors’ response to the previous reviews.

**eLife assessment**
This important study advances our understanding of how past and future information is jointly considered in visual working memory by studying gaze biases in a memory task that dissociates the locations during encoding and memory tests. The evidence supporting the conclusions is convincing, with state-of-the-art gaze analyses that build on a recent series of experiments introduced by the authors. This work, with further improvements incorporating the existing literature, will be of broad interest to vision scientists interested in the interplay of vision, eye movements, and memory.

We thank the Editors and the Reviewers for their enthusiasm and appreciation of our task, our findings, and our article. We also wish to thank the Reviewers for their constructive comments that we have embraced to improve our article. Please find below our point-by-point responses to this valuable feedback, where we also state relevant revisions that we have made to our article.

In addition, please note that we have now also made our data and code publicly available.

**Reviewer 1, Comments:**
In this study, the authors offer a fresh perspective on how visual working memory operates. They delve into the link between anticipating future events and retaining previous visual information in memory. To achieve this, the authors build upon their recent series of experiments that investigated the interplay between gaze biases and visual working memory. In this study, they introduce an innovative twist to their fundamental task. Specifically, they disentangle the location where information is initially stored from the location where it will be tested in the future. Participants are tasked with learning a novel rule that dictates how the initial storage location relates to the eventual test location. The authors leverage participants' gaze patterns as an indicator of memory selection. Intriguingly, they observe that microsaccades are directed toward both the past encoding location and the anticipated future test location. This observation is noteworthy for several reasons. Firstly, participants' gaze is biased towards the past encoding location, even though that location lacks relevance to the memory test. Secondly, there's a simultaneous occurrence of an increased gaze bias towards both the past and future locations. To explore this temporal aspect further, the authors conduct a compelling analysis that reveals the joint consideration of past and future locations during memory maintenance. Notably, microsaccades biased towards the future test location also exhibit a bias towards the past encoding location. In summary, the authors present an innovative perspective on the adaptable nature of visual working memory. They illustrate how information relevant to the future is integrated with past information to guide behavior.

Thank you for your enthusiasm for our article and findings as well as for your constructive suggestions for additional analyses that we respond to in detail below.

This short manuscript presents one experiment with straightforward analyses, clear visualizations, and a convincing interpretation. For their analysis, the authors focus on a single time window in the experimental trial (i.e., 0-1000 ms after retro cue onset). While this time window is most straightforward for the purpose of their study, other time windows are similarly interesting for characterizing the joint consideration of past and future information in memory. First, assessing the gaze biases in the delay period following the cue offset would allow the authors to determine whether the gaze bias towards the future location is sustained throughout the entire interval before the memory test onset. Presumably, the gaze bias towards the past location may not resurface during this delay period, but it is unclear how the bias towards the future location develops in that time window. Also, the disappearance of the retro cue constitutes a visual transient that may leave traces on the gaze biases which speaks again for assessing gaze biases also in the delay period following the cue offset.

Thank you for raising this important point. We initially focused on the time window during the cue given that our central focus was on gaze-biases associated with mnemonic item selection. By zooming in on this window, we could best visualize our main effects of interest: the joint selection (in time) of past and future memory attributes.

At the same time, we fully agree that examining the gaze biases over a more extended time window yields a more comprehensive view of our data. To this end, we have now also extended our analysis to include a wider time range that includes the period between cue offset (1000 ms after cue onset) and test onset (1500 ms after cue onset). We present these data below. Because we believe our future readers are likely to be interested in this as well, we have now added this complementary visualization as Supplementary Figure 4 (while preserving the focus in our main figure on the critical mnemonic selection period of interest).

**Author response image 1. sa3fig1:** Supplementary Figure 4. Gaze biases in extended time window as a complement to Figure 1 and Supplementary Figure 2. This extended analysis reveals that while the gaze bias towards the past location disappears around 600 ms after cue onset, the gaze bias towards the future location persists (panel a) and that while the early (joint) future bias occurs predominantly in the microsaccade range below 1 degree visual angle, the later bias to the future location incorporates larger eye movement that likely involve preparing for optimally perceiving the anticipated test stimulus (panel b).

This extended analysis reveals that while the gaze bias towards the past location disappears around 600 ms after cue onset (consistent with our prior reports of this bias), the gaze bias towards the future location persists. Moreover, as revealed by the data in panel b above, while the early (joint) future bias occurs predominantly in the microsaccade range below 1 degree visual angle, the later bias to the future location incorporates larger eye movement that likely involve preparing for optimally perceiving the anticipated test stimulus.

We now also call out these additional findings and figure in our article:

Page 2 (Results): “Gaze biases in both axes were driven predominantly by microsaccades (Supplementary Fig. 2) and occurred similarly in horizontal-to-vertical and vertical-tohorizontal trials (Supplementary Fig. 3). Moreover, while the past bias was relatively transient, the future bias continued to increase in anticipation of the of the test stimulus and increasingly incorporated eye-movements beyond the microsaccade range (see Supplementary Fig. 4 for a more extended time range)”.

Moreover, assessing the gaze bias before retro-cue onset allows the authors to further characterize the observed gaze biases in their study. More specifically, the authors could determine whether the future location is considered already during memory encoding and the subsequent delay period (i.e., before the onset of the retro cue). In a trial, participants encode two oriented gratings presented at opposite locations. The future rule indicates the test locations relative to the encoding locations. In their example (Figure 1a), the test locations are shifted clockwise relative to the encoding location. Thus, there are two pairs of relevant locations (each pair consists of one stimulus location and one potential test location) facing each other at opposite locations and therefore forming an axis (in the illustration the axis would go from bottom left to top right). As the future rule is already known to the participants before trial onset it is possible that participants use that information already during encoding. This could be tested by assessing whether more microsaccades are directed along the relevant axis as compared to the orthogonal axis. The authors should assess whether such a gaze bias exists already before retro cue onset and discuss the theoretical consequences for their main conclusions (e.g., is the future location only jointly used if the test location is implicitly revealed by the retro cue).

Thank you – this is another interesting point. We fully agree that additional analysis looking at the period prior to retrocue onset may also prove informative. In accordance with the suggested analysis, we have therefore now also analysed the distribution of saccade directions (including in the period from encoding to retrocue) as a function of the future rule (presented below, and now also included as Supplementary Fig. 5). Complementary recent work from our lab has shown how microsaccade directions can align to the axis of memory contents during retention (see de Vries & van Ede, eNeuro, 2024). Based on this finding, one may predict that if participants retain the items in a remapped fashion, their microsaccades may align with the axis of the future rule, and this could potentially already happen prior to cue onset.

These complementary analyses show that saccade directions are predominantly influenced by the encoding locations rather than the test locations, as seen most clearly by the saccade distribution plots in the middle row of the figure below. To obtain time-courses, we categorized saccades as occurring along the axis of the future rule or along the orthogonal axis (bottom row of the figure below). Like the distribution plots, these time course plots also did not reveal any sign of a bias along the axis of the future rule itself.

Importantly, note how this does not argue against our main findings of joint selection of past and future memory attributes, as for that central analysis we focused on saccade biases that were specific to the selected memory item, whereas the analyses we present below focus on biases in the axes in which both memory items are defined; not only the cued/selected memory item.

**Author response image 2. sa3fig2:** Supplementary Figure 5. Distribution of saccade directions relative to the future rule from encoding onset. (Top panel) The spatial layouts in the four future rules. (Middle panel) Polar distributions of saccades during 0 to 1500 ms after encoding onset (i.e. the period between encoding onset and cue onset). The purple quadrants represent the axis of the future rule and the gray quadrants the orthogonal axis. (Bottom panel) Time courses of saccades along the above two axes. We did not observe any sign of a bias along the axis of the future rule itself.

We agree that these additional results are important to bring forward when we interpret our findings. Accordingly, we now mention these findings at the relevant section in our Discussion:

Page 5 (Discussion): “First, memory contents could have directly been remapped (cf. 4,24–26) to their future-relevant location. However, in this case, one may have expected to exclusively find a future-directed gaze bias, unlike what we observed. Moreover, using a complementary analysis of saccade directions along the axis of the future rule (cf. 24), we found no direct evidence for remapping in the period between encoding and cue (Supplementary Fig. 5)”.

**Reviewer 2, Comments:**
The manuscript by Liu et al. reports a task that is designed to examine the extent to which "past" and "future" information is encoded in working memory that combines a retro cue with rules that indicate the location of an upcoming test probe. An analysis of microsaccades on a fine temporal scale shows the extent to which shifts of attention track the location of the location of the encoded item (past) and the location of the future item (test probe). The location of the encoded grating of the test probe was always on orthogonal axes (horizontal, vertical) so that biases in microsaccades could be used to track shifts of attention to one or the other axis (or mixtures of the two). The overall goal here was then to (1) create a methodology that could tease apart memory for the past and future, respectively, (2) to look at the time-course attention to past/future, and (3) to test the extent to which microsaccades might jointly encode past and future memoranda. Finally, some remarks are made about the plausibility of various accounts of working memory encoding/maintenance based on the examination of these time courses.Strengths:This research has several notable strengths. It has a clear statement of its aims, is lucidly presented, and uses a clever experimental design that neatly orthogonalizes "past" and "future" as operationalized by the authors. Figure 1b-d shows fairly clearly that saccade directions have an early peak (around 300ms) for the past and a "ramping" up of saccades moving in the forward direction. This seems to be a nice demonstration the method can measure shifts of attention at a fine temporal resolution and differentiate past from future-oriented saccades due to the orthogonal cue approach. The second analysis shown in Figure 2, reveals a dependency in saccade direction such that saccades toward the probe future were more likely also to be toward the encoded location than away from the encoded direction. This suggests saccades are jointly biased by both locations "in memory".

Thank you for your overall appreciation of our work and for highlighting the above strengths. We also thank you for your constructive comments and call for clarifications that we respond to below.

Weaknesses:(1) The "central contribution" (as the authors characterize it) is that "the brain simultaneously retains the copy of both past and future-relevant locations in working memory, and (re)activates each during mnemonic selection", and that: "... while it is not surprising that the future location is considered, it is far less trivial that both past and future attributes would be retained and (re)activated together. This is our central contribution." However, to succeed at the task, participants must retain the content (grating orientation, past) and probe location (future) in working memory during the delay period. It is true that the location of the grating is functionally irrelevant once the cue is shown, but if we assume that features of a visual object are bound in memory, it is not surprising that location information of the encoded object would bias processing as indicated by microsaccades. Here the authors claim that joint representation of past and future is "far less trivial", this needs to be evaluaed from the standpoint of prior empirical data on memory decay in such circumstances, or some reference to the time-course of the "unbinding" of features in an encoded object.

Thank you. We agree that our participants have to use the future rule – as otherwise they do not know to which test stimulus they should respond. This was a deliberate decision when designing the task. Critically, however, this does not require (nor imply) that participants have to incorporate and apply the rule to both memory items already prior to the selection cue. It is at least as conceivable that participants would initially retain the two items at their encoded (past) locations, then wait for the cue to select the target memory item, and only then consider the future location associated with the target memory item. After all, in every trial, there is only 1 relevant future location: the one associated with the cued memory item. The time-resolved nature of our gaze markers argues against such a scenario, by virtue of our observation of the joint (simultaneous) consideration of past and future memory attributes (as opposed to selection of past-before-future). These temporal dynamics are central to the insights provided by our study.

In our view, it is thus not obvious that the rule would be applied at encoding. In this sense, we do not assume that the future location is part of both memory objects from encoding, but rather ask whether this is the case – and, if so, whether the future location takes over the role of the past location, or whether past and future locations are retained jointly.

Our statements regarding what is “trivial” and what is “less trivial” regard exactly this point: it is trivial that the future is considered (after all, our task demanded it). However, it is less trivial that (1) the future location was already available at the time of initial item selection (as reflected in the simultaneous engagement of past and future locations), and (2) that in presence of the future location, the past location was still also present in the observed gaze biases.

Having said that, we agree that an interesting possibility is that participants remap both memory items to their future-relevant locations ahead of the cue, but that the past location is not yet fully “unbound” by the time of the cue. This may trigger a gaze bias not only to the new future location but also to the “sticky” (unbound) past location. We now acknowledge this possibility in our discussion (also in response to comment 3 below) where we also suggest how future work may be able to tap into this:

Page 6 (Discussion): “In our study, the past location of the memory items was technically irrelevant for the task and could thus, in principle, be dropped after encoding. One possibility is that participants remapped the two memory items to their future locations soon after encoding, and had started – but not finished – dropping the past location by the time the cue arrived. In such a scenario, the past signal is merely a residual trace of the memory items that serves no purpose but still pulls gaze. Alternatively, however, the past locations may be utilised by the brain to help individuate/separate the two memory items. Moreover, by storing items with regard to multiple spatial frames (cf. 37) – here with regard to both past and future visual locations – it is conceivable that memories may become more robust to decay and/or interference. Also, while in our task past locations were never probed, in everyday life it may be useful to remember where you last saw something before it disappeared behind an occluder. In future work, it will prove interesting to systematically vary to the delay between encoding and cue to assess whether the reliance on the past location gradually dissipates with time (consistent with dropping an irrelevant feature), or whether the past trace remains preserved despite longer delays (consistent with preserving utility for working memory).”

(2) The authors refer to "future" and "past" information in working memory and this makes sense at a surface level. However, once the retrocue is revealed, the "rule" is retrieved from long-term memory, and the feature (e.g. right/left, top/bottom) is maintained in memory like any other item representation. Consider the classic test of digit span. The digits are presented and then recalled. Are the digits of the past or future? The authors might say that one cannot know, because past and future are perfectly confounded. An alternative view is that some information in working memory is relevant and some is irrelevant. In the digit span task, all the digits are relevant. Relevant information is relevant precisely because it is thought be necessary in the future. Irrelevant information is irrelevant precisely because it is not thought to be needed in the immediate future. In the current study, the orientation of the grating is relevant, but its location is irrelevant; and the location of the test probe is also relevant.

Thank you for this stimulating reflection. We agree that in our set-up, past location is technically “task-irrelevant” while future location is certainly “task-relevant”. At the same time, the engagement of the past location suggests to us that the brain uses past location for the selection – presumably because the brain uses spatial location to help individuate/separate the items, even if encoded locations are never asked about. Therefore, whether something is relevant or irrelevant ultimately depends on how one defines relevance (past location may be relevant/useful for the brain even if technically irrelevant from the perspective of the task). In comparison, the use of “past” and “future” may be less ambiguous.

It is also worth noting how we interpret our findings in relation to demands on visual working memory, inspired by dynamic situations whereby visual stimuli may be last seen at one location but expected to re-appear at another, such as a bird disappearing behind a building (the example in our introduction). Thus, past for us does not refer to the memory item perse (like in the digit span analogue) but, rather, quite specifically to the past location of a dynamic visual stimulus in memory (which, in our experiment, was operationalised by the future rule, for convenience).

(3) It is not clear how the authors interpret the "joint representation" of past and future. Put aside "future" and "past" for a moment. If there are two elements in memory, both of which are associated with spatial bindings, the attentional focus might be a spatial average of the associated spatial indices. One might also view this as an interference effect, such that the location of the encoded location attracts spatial attention since it has not been fully deleted/removed from working memory. Again, for the impact of the encoded location to be exactly zero after the retrieval cue, requires zero interference or instantaneous decay of the bound location information. It would be helpful for the authors to expand their discussion to further explain how the results fit within a broader theoretical framework and how it fits with empirical data on how quickly an irrelevant feature of an object can be deleted from working memory.

Thank you also for this point (that is related to the two points above). As we stated in our reply to comment 1 above, we agree that one possibility is that the past location is merely “sticky” and pulls the task-relevant future bias toward the past location. If so, our time courses suggest that such “pulling” occurs only until approximately 600 ms after cue onset, as the past bias is only transient. An alternative interpretation is that the past location may not be merely a residual irrelevant trace, but actually be useful and used by the brain.

For example, the encoded (past) item locations provide a coordinate system in which to individuate/separate the two memory items. While the future locations also provide such a coordinate system, the brain may benefit from holding onto both coordinate systems at the same time, rendering our observation of joint selection in both frames. Indeed, in a recent VR experiment in which we had participants (rather than the items) rotate, we also found evidence for the joint use of two spatial frames, even if neither was technically required for the upcoming task (see Draschkow, Nobre, van Ede, Nature Human Behaviour, 2022). Though highly speculative at this stage, such reliance on multiple spatial frames may make our memories more robust to decay and/or interference. Moreover, while past location was never explicitly probed in our task, in daily life the past location may sometimes (unexpectedly) become relevant, hence it may be useful to hold onto it, just in case. Thus, considering the past location merely as an “irrelevant feature” (that takes time to delete) may not do sufficient justice to the potential roles of retaining past locations of dynamic visual objects held in working memory.

As also stated in response to comment 1 above, we now added these relevant considerations to our Discussion:

Page 5 (Discussion): “In our study, the past location of the memory items was technically irrelevant for the task and could thus, in principle, be dropped after encoding. One possibility is that participants remapped the two memory items to their future locations soon after encoding, and had started – but not finished – dropping the past location by the time the cue arrived. In such a scenario, the past signal is merely a residual trace of the memory items that serves no purpose but still pulls gaze. Alternatively, however, the past locations may be utilised by the brain to help individuate/separate the two memory items. Moreover, by storing items with regard to multiple spatial frames (cf. 37) – here with regard to both past and future visual locations – it is conceivable that memories may become more robust to decay and/or interference. Also, while in our task past locations were never probed, in everyday life it may be useful to remember where you last saw something before it disappeared behind an occluder. In future work, it will prove interesting to systematically vary to the delay between encoding and cue to assess whether the reliance on the past location gradually dissipates with time (consistent with dropping an irrelevant feature), or whether the past trace remains preserved despite longer delays (consistent with preserving utility for working memory).”

**Reviewer 3, Comments:**
This study utilizes saccade metrics to explore, what the authors term the "past and future" of working memory. The study features an original design: in each trial, two pairs of stimuli are presented, first a vertical pair and then a horizontal one. Between these two pairs comes the cue that points the participant to one target of the first pair and another of the second pair. The task is to compare the two cued targets. The design is novel and original but it can be split into two known tasks - the first is a classic working memory task (a post-cue informs participants which of two memorized items is the target), which the authors have used before; and the second is a classic spatial attention task (a pre-cue signal that attention should be oriented left or right), which was used by numerous other studies in the past. The combination of these two tasks in one design is novel and important, as it enables the examination of the dynamics and overlapping processes of these tasks, and this has a lot of merit. However, each task separately is not new. There are quite a few studies on working memory and microsaccades and many on spatial attention and microsaccades. I am concerned that the interpretation of "past vs. future" could mislead readers to think that this is a new field of research, when in fact it is the (nice) extension of an existing one. Since there are so many studies that examined pre-cues and post-cues relative to microsaccades, I expected the interpretation here to rely more heavily on the existing knowledge base in this field. I believe this would have provided a better context of these findings, which are not only on "past" vs. "future" but also on "working memory" vs. "spatial attention".

Thank you for considering our findings novel and important, while at the same time reminding us of the parallels to prior tasks studying spatial attention in perception and working memory. We fully agree that our task likely engages both attention to the (past) memory item as well as spatial attention to the upcoming (future) test stimulus. At the same time, there is a critical difference in spatial attention for the future in our task compared with ample prior tasks engaging spatial cueing of attention for perception. In our task, the cue never directly cues the future location. Rather, it exclusively cues the relevant memory item. It is the memory item that is associated with the relevant future location, according to the future rule. This integration of the rule-based future location into the memory representation is distinct from classical spatial-attention tasks in which attention is cued directly to a specific location via, for example, a spatial cue such as an arrow.

Thus, if we wish to think about our task as engaging cueing of spatial attention for perception, we have to at least also invoke the process of cueing the relevant location via the appropriate memory item. We feel it is more parsimonious to think of this as attending to both the past and future location of a dynamic visual object in working memory.

If we return to our opening example, when we see a bird disappear behind a building, we can keep in working memory where we last saw it, while anticipating where it will re-appear to guide our external spatial attention. Here too, spatial attention is fully dependent on working-memory content (the bird itself) – mirroring the dynamic semng in our study. Thus, we believe our findings contribute a fresh perspective, while of course also extending established fields. We now contextualize our finding within the literature and clarify our unique contribution in our revised manuscript:

Page 5 (Discussion): “Building on the above, at face value, our task may appear like a study that simply combines two established tasks: tasks using retro-cues to study attention in working memory (e.g.,2,31-33) and tasks using pre-cues to study orienting of spatial attention to an upcoming external stimulus (e.g., 31,32,34–36). A critical difference with common pre-cue studies, however, is that the cue in our task never directly informed the relevant future location. Rather, as also stressed above, the future location was a feature of the cued memory item (according to the future rule), and not of the cue itself. Note how this type of scenario may not be uncommon in everyday life, such as in our opening example of a bird flying behind a building. Here too, the future relevant location is determined by the bird – i.e. the memory content – itself.”

**Reviewer 2, Recommendations:**
It would be helpful to set up predictions based on existing working memory models. Otherwise, the claim that the joint coding of past/future is "not trivial" is simply asserted, rather than contradicting an existing model or prior empirical results. If the non-trivial aspect is simply the ability to demonstrate the joint coding empirical through a good experimental design, make it clear that this is the contribution. For example, it may be that prevailing models predict exactly this finding, but nobody has been able to demonstrate it cleanly, as the authors do here. So the non-triviality is not that the result contradicts working memory models, but rather relates to the methodological difficulty of revealing such an effect.

Thank you for your recommendation. First, please see our point-by-point responses to the individual comments above, where we also state relevant changes that we have made to our article, and where we clarify what we meant with “non trivial”. As we currently also state in our introduction, our work took as a starting point the framework that working memory is inherently about the past while being for the future (cf. van Ede & Nobre, Annual Review of Psychology, 2023). By virtue of our unique task design, we were able to empirically demonstrate that visual contents in working memory are selected via both their past and their future-relevant locations – with past and future memory attributes being engaged together in time. With “not trivial” we merely intend to make clear that there are viable alternatives than the findings we observed. For example, past could have been replaced by the future, or it could have been that item selection (through its past location) was required before its future-relevant location could be considered (i.e. past-before-future, rather than joint selection as we reported). We outline these alternatives in the second paragraph of our Discussion:

Page 5 (Discussion): “Our finding of joint utilisation of past and future memory attributes emerged from at least two alternative scenarios of how the brain may deal with dynamic everyday working memory demands in which memory content is encoded at one location but needed at another.

First, [….]”

Our work was not motivated from a particular theoretical debate and did not aim to challenge ongoing debates in the working-memory literature, such as: slot vs. resource, active vs. silent coding, decay vs. interference, and so on. To our knowledge, none of these debates makes specific claims about the retention and selection of past and future visual memory attributes – despite this being an important question for understanding working memory in dynamics everyday semngs, as we hoped to make clear by our opening example.

**Reviewer 3, Recommendations:**
I recommend that the present findings be more clearly interpreted in the context of previous findings on working memory and attention. The task design includes two components - the first (post-cue) is a classic working memory task and the second (the pre-cue) is a classic spatial attention design. Both components were thoroughly studied in the past and this previous knowledge should be better integrated into the present conclusions. I specifically feel uncomfortable with the interpretation of past vs. future. I find this framework to be misleading because it reads like this paper is on a topic that is completely new and never studied before, when in fact this is a study on the interaction between working memory and spatial attention. I recommend the authors minimize this past-future framing or be more explicit in explaining how this new framework relates to the more common terminology in the field and make sure that the findings are not presented in a vacuum, as another contribution to the vibrant field that they are part of.

Thank you for these recommendations. Please also see our point-by-point responses to the individual comments above. Here, we explained our logic behind using the terminology of past vs. future (in addition, see also our response to point 2 or reviewer 2). Here, we also stated relevant changes that we have made to our manuscript to explain how our findings complement – but are also distinct from – prior tasks that used pre-cues to direct spatial attention to an upcoming stimulus. As we explained above, in our task, the cue itself never contained information about the upcoming test location. Rather, the upcoming test location was a property of the memory item (given the future rule). Hence, we referred to this as a “future attribute” of the cued memory item, rather than as the “cued location” for external spatial attention. Still, we agree the future bias likely (also) reflects spatial allocation to the upcoming test array, and we explicitly acknowledge this in our discussion. For example:

Page 5 (Discussion): “This signal may reflect either of two situations: the selection of a future-copy of the cued memory content or anticipatory attention to its the anticipated location of its associated test-stimulus. Either way, by the nature of our experimental design, this future signal should be considered a content-specific memory attribute for two reasons. First, the two memory contents were always associated with opposite testing locations, hence the observed bias to the relevant future location must be attributed specifically to the cued memory content. Second, we cued which memory item would become tested based on its colour, but the to-be-tested location was dependent on the item’s encoding location, regardless of its colour. Hence, consideration of the item’s future-relevant location must have been mediated by selecting the memory item itself, as it could not have proceeded via cue colour directly.”

Page 6 (Discussion): “Building on the above, at face value, our task may appear like a study that simply combines two established tasks: tasks using retro-cues to study attention in working memory (e.g.,2,31-33) and tasks using pre-cues to study orienting of spatial attention to an upcoming external stimulus (e.g., 31,32,34–36). A critical difference with common pre-cue studies, however, is that the cue in our task never directly informed the relevant future location. Rather, as also stressed above, the future location was a feature of the cued memory item (according to the future rule), and not of the cue itself. Note how this type of scenario may not be uncommon in everyday life, such as in our opening example of a bird flying behind a building. Here too, the future relevant location is determined by the bird – i.e. the memory content – itself.”